# Quality Assessment of HDR/WCG Images Using HDR Uniform Color Spaces

**DOI:** 10.3390/jimaging5010018

**Published:** 2019-01-14

**Authors:** Maxime Rousselot, Olivier Le Meur, Rémi Cozot, Xavier Ducloux

**Affiliations:** 1Harmonic Inc., ZAC des Champs Blancs, 57 Rue Clément Ader, 35510 Cesson-Sévigné, France; 2Institut de Recherche en Informatique et Systèmes Aléatoires (IRISA), Campus Universitaire de Beaulieu, 35042 Rennes CEDEX, France

**Keywords:** image quality, High Dynamic Range, color gamut, image representation, image analysis, image processing

## Abstract

High Dynamic Range (HDR) and Wide Color Gamut (WCG) screens are able to render brighter and darker pixels with more vivid colors than ever. To assess the quality of images and videos displayed on these screens, new quality assessment metrics adapted to this new content are required. Because most SDR metrics assume that the representation of images is perceptually uniform, we study the impact of three uniform color spaces developed specifically for HDR and WCG images, namely, ICtCp, $J_{z}a_{z}b_{z}$ and HDR-Lab on 12 SDR quality assessment metrics. Moreover, as the existing databases of images annotated with subjective scores are using a standard gamut, two new HDR databases using WCG are proposed. Results show that MS-SSIM and FSIM are among the most reliable metrics. This study also highlights the fact that the diffuse white of HDR images plays an important role when adapting SDR metrics for HDR content. Moreover, the adapted SDR metrics does not perform well to predict the impact of chrominance distortions.

## 1. Introduction

Screen technologies have made incredible progress in recent years. They are able to display brighter and darker pixels with more vivid colors than ever and, thus, create more impressive and realistic images.

Indeed, the new generation of screens can display a luminance that can go below 0.01 ${\mathrm{cd}/m}^{2}$ and up to 10,000 ${\mathrm{cd}/m}^{2}$, thus, allowing them to handle images and video with a High Dynamic Range (HDR) of luminance. For comparison, screens with a standard dynamic range (SDR) are traditionally able to display luminance between 1 and 100 ${\mathrm{cd}/m}^{2}$. To handle HDR images, new transfer functions have to be used to transform the true linear light to perceptually linear light (Opto-Electronic Transfer Function (OETF)). The function used to transform back the perceptually linear light to true linear light is called the Electro-Optic Transfer Function (EOTF). The OETF and the EOTF functions are not exactly the inverse of each other. This non-linearity compensates the differences in tonal perception between the environment of the camera and that of the display. The SDR legacy transfer functions called gamma functions are normalized in BT.709 [1] and BT.1886 [2]. For HDR video compression, two transfer functions were standardized: PQ [3] (Perceptual Quantizer) and HLG (Hybrid-Log-Gamma) [4].

Screen enhancements do not only focus on increasing the luminance range but also on the size of the color space it can cover. Indeed, the color space that a screen can display is limited by the chromatic coordinates of its three primary colors (Red, Green and Blue) corresponding to the three kinds of display photo-transmitters. The gamut, i.e., a subset of visible colors that can be represented by a color space, used to encode SDR images (normalized in BT.709 [1]) is not wide enough to cover the gamut that could be displayed by a Wide Color Gamut (WCG) screen. The BT.2020 [5] recommendation define how to handle this wider gamut. For the moment, there is no screen that can cover this gamut in its totality, but some are really close. The standard BT.2100 [6] sums up all the aforementioned HDR/WCG standards.

For these new images and videos, new quality assessment metrics are required. Indeed, quality metrics are key tools to assess performances of diverse image processing applications such as image and video compression. Unfortunately, SDR image quality metrics are not appropriate for HDR/WCG contents. To overcome this problem, we can follow two strategies. The first one is to adapt existing SDR metrics to a higher dynamic range. For instance, instead of using a classical gamma transfer function, Aydin et al. [7] defined a transfer function, called the Perceptually Uniform (PU) function, which corresponds to the gamma non-linearity (defined in BT.1886 [2]) for luminance value between 0.1 and 80 ${\mathrm{cd}/m}^{2}$ while retaining perceptual linearity above. This method can be used for any metrics using the luminance corrected with the gamma function, such as PSNR, SSIM [8], VIF [9], Multiscale-SSIM [10] (MS-SSIM). In this paper, the metrics using the Perceptually Uniform (PU) function have the prefix PU- (PU-PSNR, PU-SSIM). The second strategy is to design dedicated metrics for HDR contents. We can mention HDR-VDP2 [11,12] for still images and HDR-VQM [13] for videos.

Several studies have already benchmarked the performances of HDR metrics. In [14], the authors assessed the performances of 35 quality metrics over 240 HDR images compressed with JPEG XT [15]. They conclude that HDR-VDP2 (version 2.2.1 [12]) and HDR-VQM were the best performing metrics, closely followed by PU-MS-SSIM. In [16], the authors came to the conclusion that HDR-VDP2 (version 2.1.1) can be successfully used for predicting the quality of video pair comparison contrary to HDR-VQM. In [17], authors showed that HDR-VDP2, HDR-VQM, PU-VIF and PU-SSIM provide similar performances. In [18], results indicate that PU-VIF and HDR-VDP2 have similar performances, although PU-VIF has a slightly better reliability than HDR-VDP2 for lower quality scores. More recently, Zerman et al. [19] demonstrated that HDR-VQM is the best full-reference HDR quality metric.

The above studies have two major limitations. First, as all of these metrics are color-blind, they only provide an answer to the increase of luminance range but they do not consider the WCG gamut. Second, the databases used to evaluate the different metrics were most of the time defined with an HDR display only capable of displaying a BT.709 [1] gamut. The WCG gamut BT.2020 [5] is currently addressed neither by current metrics nor by current databases.

To overcome these limitations, in this paper, we adapt existing SDR metrics to HDR/WCG images using uniform color spaces adapted to HDR. Indeed, most SDR metrics assume that the representation of images is perceptually linear. To be able to evaluate HDR metrics that include both luminance and chromatic information, we propose two new image databases, that include chrominance artifacts within the BT.2020 wide color gamut.

This paper is organized as follows. First, we describe the adaptation of SDR metrics to HDR/WCG images using perceptually uniform color spaces. Second, we present the methodology used to evaluate the performances of these metrics. In a third part, the performances of the considered metrics are presented. Results are discussed in a fourth part. A fifth section describes our recommendation to assess the quality of HDR/WCG images. The last section concludes this paper.

## 2. From State-of-the-Art SDR Quality Assessment Metrics to HDR/WCG Quality Assessment Metrics

In this section we first present the perceptually uniform color spaces able to encode the HDR/WCG content. Then in a second part, we elaborate on the color difference metrics associated with these color space. In a third part, we describe a selection of SDR quality metrics. Finally, we present how we tailor SDR quality metrics to HDR/WCG content.

### 2.1. Perceptually Uniform Color Spaces

For many image processing applications such as compression and quality assessment, pixels are encoded with a three-dimensional representation: one dimension corresponds to an achromatic component the luminance and the two others correspond to the chromatic information. An example of this kind of representations is the linear color space CIE-XYZ where *Y* represents the luminance and *X* and *Z* the chromatic information. This color space is often used as a reference from which many other color spaces are derived (including most of RGB color spaces). However this space is not a uniform (or perceptually uniform) color space. A uniform color space is defined so that the difference between two values always corresponds to the same amount of visually perceived change.

Three uniform color spaces are considered in this article: HDR-Lab [20], the HDR extension of the CIE 1976 $L^{*}a^{*}b^{*}$ [21] and two other HDR/WCG color spaces designed to be perceptually linear, and simple to use: ICtCp [6] and $J_{z}a_{z}b_{z}$ [22]. Unlike the XYZ color space in which all components are always non-negative, these three uniform color spaces represent the chromatic information using a color-opponent model which is coherent with the Human Visual System (HVS) and the opponent color theory.

In this article, the luminance component of the uniform color spaces is called uniform luminance instead of, according to the case, lightness, brightness or luma to avoid unnecessary complexity. For example, the uniform luminance of HDR-Lab should be called lightness while the uniform luminance of $J_{z}a_{z}b_{z}$ should be called brightness.

#### 2.1.1. HDR-Lab

One of the most popular uniform color spaces is the CIE 1976 $L^{*}a^{*}b^{*}$ or CIELAB which is suited for SDR content. An extension of this color space for HDR images was proposed in [20]. The proposition is to tailor CIELAB for HDR by changing the non-linear function applied to the pixel XYZ values. This color space is calculated as follows:(1)$$\left\{\begin{array}{ll} L_{HDR} & =f(Y/Y_{n})\\ a_{HDR} & =5[f\left(X/X_{n}\right)-f\left(Y/Y_{n}\right)]\\ b_{HDR} & =2[f\left(Y/Y_{n}\right)-f\left(Z/Z_{n}\right)] \end{array}\right.$$
where $X_{n}$, $Y_{n}$ and $Z_{n}$ are the XYZ coordinates of the diffuse white. The non-linear function *f* is used to output perceptually linear values. *f* is defined for HDR as follows:(2)$$f\left(\omega \right)=247\frac{\omega^{\epsilon }}{\omega^{\epsilon }+2^{\epsilon }}+0.02$$
(3)ϵ=0.58/(sf×lf)
(4)$$sf=1.25-0.25(Y_{s}/0.184)$$
(5)$$lf=\log\left(318\right)/\log\left(Y_{n}\right)$$
where $Y_{S}$ is the relative luminance of the surround and $Y_{n}$ is the absolute luminance of the diffuse white or reference white. The diffuse white corresponds to the chromatic coordinates in the XYZ domain of a 100% reflectance white card without any specular highlight. In HDR imaging, the luminance *Y* of the diffuse white is different from the luminance of the peak brightness. Light coming from specular reflections or emissive light sources can reach much higher luminance values. The luminance of the diffuse white is often chosen during the color grading of the images.

The use of HDR-Lab color space is somewhat difficult since it requires to know the relative luminance of the surround, $Y_{S}$, as well as the diffuse white luminance, $Y_{n}$. Unfortunately these two parameters are most of the time unknown for HDR contents. To cope with this issue, we consider two different diffuse whites to compute the HDR-Lab color space, i.e., 100 ${\mathrm{cd}/m}^{2}$ and 1000 ${\mathrm{cd}/m}^{2}$. These two color spaces are named $HDR\text{-}Lab_{100}$ and $HDR\text{-}Lab_{1000}$, respectively.

In addition to the HDR-Lab color space, Fairchild et al. [20] also proposed the HDR-IPT color space, which aims to extent the IPT color space [23] to HDR content. This color space is not studied in this article due to its high similarity with HDR-Lab.

#### 2.1.2. ICtCp

ICtCp has a better chrominance and luminance decorrelation and has a better hue linearity than the classical $Y^{\prime }CrCb$ color space [24]. This color space is calculated in three steps:First, the linear RGB values (in the BT.2020 gamut) are converted into LMS values which correspond to the quantity of light absorbed by the cones:
(6)$$\left\{\begin{array}{lllll} L & = & 0.41210938\times R & \;+0.52392578\times G & \;+0.06396484\times B\\ M & = & 0.16674805\times R & \;+0.72045898\times G & \;+0.11279297\times B\\ S & = & 0.02416992\times R & \;+0.07543945\times G & \;+0.90039062\times B \end{array}\right.$$Second, the inverse EOTF PQ [6] is applied to each LMS component:
(7)$$\left\{\begin{array}{ccc} L^{\prime } & = & \;EOTF_{PQ}^{-1}\left(L\right)\\ M^{\prime } & = & \;EOTF_{PQ}^{-1}\left(M\right)\\ S^{\prime } & = & \;EOTF_{PQ}^{-1}\left(S\right) \end{array}\right.$$Finally, the luminance component *I* and the chrominance components Ct and Cp are deduced as follows:
(8)$$\left\{\begin{array}{ccccc} I & = & 0.5\times L^{\prime } & \;+0.5\times M^{\prime }\\ Ct & = & 1.61376953\times L^{\prime } & \;-3.32348632\times M^{\prime } & \;+1.70971679\times S^{\prime }\\ Cp & = & 4.37817382\times L^{\prime } & \;-4.37817383\times M^{\prime } & \;-0.13256835\times S^{\prime } \end{array}\right.$$

The ICtCp color space [6] is particularly well adapted to video compression and more importantly to the PQ EOTF as defined in BT.2100 [6].

#### 2.1.3. $J_{z}a_{z}b_{z}$

$J_{z}a_{z}b_{z}$ [22] is a uniform color space allowing to increase the hue uniformity and to predict accurately small and large color differences, while keeping a low computational cost. It is computed from the XYZ values (with a standard illuminant D65) in five steps:First, the XYZ values are adjusted to remove a deviation in the blue hue.
(9)$$\left[\begin{array}{c} X^{\prime }\\ Y^{\prime } \end{array}\right]=\left[\begin{array}{c} bX\\ gY \end{array}\right]-\left[\begin{array}{c} (b-1)Z\\ (g-1)X \end{array}\right]$$
where b=1.15 and g=0.66.Second, the $X^{\prime }Y^{\prime }Z$ values are converted to LMS values
(10)$$\left\{\begin{array}{ccccc} L & = & 0.41478972\times X^{\prime } & \;+0.579999\times Y^{\prime } & \;+0.0146480\times Z\\ M & = & -0.2015100\times X^{\prime } & \;+1.120649\times Y^{\prime } & \;+0.0531008\times Z\\ S & = & -0.0166008\times X^{\prime } & \;+0.264800\times Y^{\prime } & \;+0.6684799\times Z \end{array}\right.$$Third, as for ICtCp, the inverse EOTF PQ is applied on each LMS component
(11)$$\left\{\begin{array}{ccc} L^{\prime } & = & \;EOTF_{PQ}^{-1}\left(L\right)\\ M^{\prime } & = & \;EOTF_{PQ}^{-1}\left(M\right)\\ S^{\prime } & = & \;EOTF_{PQ}^{-1}\left(S\right) \end{array}\right.$$Fourth, the luminance $I_{z}$ and the chrominance $a_{z}$ and $b_{z}$ are calculated
(12)$$\left\{\begin{array}{ccccc} I_{z} & = & 0.5\times L^{\prime } & \;+0.5\times M^{\prime }\\ b_{z} & = & 3.5240000\times L^{\prime } & \;-4.0667080\times M^{\prime } & \;+0.5427080\times S^{\prime }\\ a_{z} & = & 0.1990776\times L^{\prime } & \;+1.0967990\times M^{\prime } & \;-1.2958750\times S^{\prime } \end{array}\right.$$Finally, to handle the highlight, the luminance is adjusted:
(13)$$J_{z}=\frac{\left(1+d\right)\times I_{z}}{1+(d\times I_{z})}-d_{0}$$
where $J_{z}$ is the adjusted luminance, d=−0.56 and $d_{0}$ is a small constant: $d_{0}=1.6295499532812566\times 10^{-11}$.

### 2.2. Color Difference Metrics

In this section, we present the color difference metrics associated to each HDR color space. Because the color spaces are uniform, it is possible to calculate the perceptual difference between two colors.
For HDR-Lab color space, the Euclidean distance $\Delta E_{HDR\text{-}Lab}$ is used:
(14)$$\Delta E_{HDR\text{-}Lab}=\sqrt{{\left(\Delta L\right)}^{2}+{\left(\Delta a\right)}^{2}+{\left(\Delta b\right)}^{2}}$$For the $J_{z}a_{z}b_{z}$, Safdar et al. [22] proposed the following formula:
(15)$$\Delta E_{J_{z}a_{z}b_{z}}=\sqrt{{\left(\Delta J_{z}\right)}^{2}+{\left(\Delta C_{z}\right)}^{2}+{\left(\Delta H_{z}\right)}^{2}}$$
where $C_{z}$ corresponds to the color saturation and $h_{z}$ to the hue:
(16)$$C_{z}=\sqrt{{\left(a_{z}\right)}^{2}+{\left(b_{z}\right)}^{2}}$$
(17)$$h_{z}=arctan\left(\frac{b_{z}}{a_{z}}\right)$$
(18)$$\Delta H_{z}=2\sqrt{C_{z1}C_{z2}}\times sin\left(\frac{\Delta h_{z}}{2}\right)$$
where $C_{z1}$ and $C_{z2}$ correspond to the saturation of the two compared colors.For ICtCp, a weighted Euclidean distance formula was proposed in [25]:
(19)$$\Delta E_{ICtCp}=720\sqrt{{\left(\Delta I\right)}^{2}+0.25{\left(\Delta Ct\right)}^{2}+{\left(\Delta Cp\right)}^{2}}$$
Then, to have a ICtCp color space truly perceptually linear, the coefficient 0.25 is applied to the Ct component before using any SDR metric.

These color difference metrics work well for measuring perceptual differences of uniform patches. Although that we do not perceive color differences in the same way in textured images or in uniform and large patches, they are often used to compare the distortion between two images. The mean of the difference between the distorted and the reference images can be used as an indicator of image quality:(20)$${\bar{\Delta E}}_{colorspace}=\frac{1}{IJ}\sum_{i=1}^{I}\sum_{j=1}^{J}\Delta E_{colorspace}\left(i,j\right)$$
where *I* and *J* correspond to the dimensions of the image and (i,j) corresponds to the spatial coordinates of the pixel.

### 2.3. SDR Quality Assessment Metrics

We have selected 12 SDR metrics commonly used in academic research, standardization or industry. There are six achromatic or color-blind metrics (PSNR, SSIM, MS-SSIM, FSIM, PSNR-HVS-M and PSNR-HMA) and six metrics including chrominance information ($\bar{\Delta E}$, $\bar{\Delta E^{S}}$, SSIMc, CSSIM, FSIMc, PSNR-HMAc). Table 1 summarizes the principle of each metrics. More detailed information about these metrics can be found in a Appendix D.

### 2.4. Adapting SDR Metrics to HDR/WCG Images

For adapting SDR metrics to HDR/WCG images, the reference and distorted images are first converted in a perceptually linear color space. A remapping function is then applied. Finally the SDR metrics is used to determine the quality score. Figure 1 presents the diagram of the proposed method.

#### 2.4.1. Color Space Conversion

Most SDR metrics were designed with the assumption that the images are encoded in the legacy $Y^{\prime }Cr^{\prime }Cb^{\prime }$ [1] color space; this color space is approximately perceptually uniform for SDR content.

To use SDR metrics with HDR images, we propose to leverage perceptually uniform color spaces adapted to HDR and WCG images (ICtCp, $J_{z}a_{z}b_{z}$, $HDR\text{-}Lab_{100}$ and $HDR\text{-}Lab_{1000}$).

To illustrate the importance of using uniform color space, we also consider two non-uniform color spaces, namely XYZ and $Y^{\prime }Cr^{\prime }Cb^{\prime }$ color spaces as defined in the BT.2020 recommendation [5]. This last space can’t be considered as approximatly uniform for HDR content as it uses the classical gamma function. This function is applied to the RGB component of an image:(21)$$E^{\prime }=\left\{\begin{array}{ccc} 4.5E & \mathrm{if} & 0\leq E\leq \beta \\ \alpha E^{0.45}-\left(\alpha -1\right) & \mathrm{otherwise} \end{array}\right.$$
where α=1.099, β=0.018 and *E* is one of the *R*, *G* and *B* channel normalized by the reference white level. In SDR, it is supposed to be equal to the peak brightness of the display, so we choose as being the maximum value taken by our own HDR images: 4250 ${\mathrm{cd}/m}^{2}$.

From the non-linear $R^{\prime }B^{\prime }G^{\prime }$ color space, the $Y^{\prime }Cr^{\prime }Cb^{\prime }$ color space can be easily deduced:(22)$$\left\{\begin{array}{ccc} Y^{\prime } & = & 0.2627\times R^{\prime }+0.6780\times G^{\prime }+0.0593\times B^{\prime }\\ Cr^{\prime } & = & (R^{\prime }-Y^{\prime })/1.4746\\ Cb^{\prime } & = & (B^{\prime }-Y^{\prime })/1.8814 \end{array}\right.$$

In addition to the previous color space, for the color-blind metrics, we use the PU-mapping function for the luminance [7]. As mentioned earlier, this transfer function keeps the same behaviour than the $Y^{\prime }Cr^{\prime }Cb^{\prime }$ with a reference white of 80 ${\mathrm{cd}/m}^{2}$ (which is perceptually linear within a SDR range) and retains perceptual linearity above. Thus any color-blind metrics can be used thanks to this mapping.

#### 2.4.2. Remapping Function

The six aforementioned color spaces, i.e., XYZ, $Y^{\prime }Cr^{\prime }Cb^{\prime }$, $HDR\text{-}Lab_{100}$, $HDR\text{-}Lab_{1000}$, ICtCp and $J_{z}a_{z}b_{z}$, have a different range of values. As most of SDR metrics have constant values defined for pixel values between 0 and 255, it is required to adapt the color spaces. We remap them in a way that their respective perceptually linear luminances fit a similar range as the luminances encoded with the PU transfer function between 0 and 100 ${\mathrm{cd}/m}^{2}$. We choose 100 ${\mathrm{cd}/m}^{2}$ as a normalization point because it roughly corresponds to the peak brightness of an SDR screen. Moreover, the PU-encoding is used as a reference to remap the color spaces because it is already adapted to SDR metrics. The goal of this process is to obtain HDR images with the same luminance scale than SDR images in the range 0 to 100 ${\mathrm{cd}/m}^{2}$ while preserving the perceptual uniformity of the color spaces. The remapping of the perceptual color spaces is done as follows:
(23)$$\hat{J_{z}a_{z}b_{z}}\left(i,j\right)=\frac{\alpha_{PU}}{\beta_{J_{z}}}\times J_{z}a_{z}b_{z}\left(i,j\right)$$
where $J_{z}a_{z}b_{z}\left(i,j\right)$ corresponds to the value in the $J_{z}a_{z}b_{z}$ domain of the pixel with the spatial coordinates *i* and *j*. $\hat{J_{z}a_{z}b_{z}}\left(i,j\right)$ corresponds to the same pixel value after the remapping. $\alpha_{PU}$ is the luminance value in the PU space when linear luminance value is 100 ${\mathrm{cd}/m}^{2}$. $\beta_{J_{z}}$ is the same value but for the luminance component $J_{z}$ of the $J_{z}a_{z}b_{z}$ color space. A similar operation is applied to ICtCp and HDR-Lab, XYZ and $Y^{\prime }Cr^{\prime }Cb^{\prime }$ color spaces. The resulting luminances for the aforementioned color-space as well as the PU-encoding luminance are plot on Figure 2. For these figures, we chose a surround luminance of 20 ${\mathrm{cd}/m}^{2}$ for the two HDR-Lab color spaces.

**Remark** **1.**

*Note that, to adapt the $\bar{\Delta E^{S}}$ metrics, the blurring model used in this metrics is first applied to the XYZ color space of the images and then the different color difference metrics are calculated. In the case of the $J_{z}a_{z}b_{z}$ color space instead of the color difference metrics presented in Equation (15), we use a simpler Euclidean distance between the pixel values.*

*In the following sections, the naming convention used for all metrics is ${Metrics}_{ColorSpace}$. For example, the PSNR metrics used with the ICtCp color space is called ${PSNR}_{ICtCp}$.*



## 3. Methodology for the Quality Assessment Metrics Evaluation

In this section, we describe the methodology used to evaluate the performances of the adapted metrics described in the previous section. First, we present the HDR image databases annotated with subjective score or Mean Opinion Score (MOS). They are used to compare the objective metrics quality scores to a ground truth. Then, we present the performance indexes used to perform this comparison.

### 3.1. Databases Presentation

Five image databases are used for carrying out the comparison. Three were already available online and two were created to handle WCG image quality assessment. To describe objectively the images of each database, we use four indicators:the image dynamic range (DR):
(24)$$\mathrm{DR}=\log\left(Y_{max}\right)-\log\left(Y_{min}\right)$$
where $Y_{min}$ and $Y_{max}$ are the minimum and the maximum luminance (in the XYZ linear domain), respectively. They are computed after excluding 1% of the brightest and darkest pixels to be more robust to outliers.the key of the picture is a measure of its overall brightness (in the range [0, 1]):
(25)$$\mathrm{key}=\frac{\bar{\log(Y)}-\log\left(Y_{min}\right)}{\log\left(Y_{max}\right)-\log\left(Y_{min}\right)}$$
$\bar{\log(Y)}$ is computed as follows:
(26)$$\bar{\log(Y)}=\frac{1}{N}\sum_{i=1}^{I}\sum_{j=1}^{J}\log\left(Y\left(i,j\right)+\delta \right)$$
where Y(i,j) is the luminance of pixel (i,j), *N* the total number of pixels and δ a small offset to avoid a singularity when the luminance is null. $Y_{min}$ and $Y_{max}$ are calculated as for the dynamic range.the spatial information (SI) [32] describes the complexity of an image. It corresponds to the standard deviation of the image luminance plane which has been filtered by a Sobel filter:
(27)SI=std[Sobel(Y)]
On SDR images, this metrics is used after the OETF, usually a gamma function, and, thus, making the luminance approximately perceptually linear. To be as similar as values in SDR, the PU function is applied on the luminance of the HDR images.the colorfulness is a metrics of the perceived saturation of an image [33]. The ${\hat{M}}^{\left(1\right)}$ version of the metrics is used because the image is first converted in the CIELab space, a space that can be adapted to HDR (cf. Section 2.1). This metrics is computed as follows:
(28)$${\hat{M}}^{\left(1\right)}=\sigma_{ab}+0.37\times \mu_{ab}$$
where
(29)$$\sigma_{ab}=\sqrt{\sigma_{a}^{2}+\sigma_{b}^{2}}$$
and
(30)$$\mu_{ab}=\sqrt{\mu_{a}^{2}+\mu_{b}^{2}}$$
where $\sigma_{a}$ and $\sigma_{b}$ are the standard deviations along the *a* and *b* axis, respectively. $\mu_{a}$ and $\mu_{b}$ are the means of the *a* and *b* component, respectively.

Before calculating these indicators, the image luminance is limited to the display available dynamic range used in the respective subjective tests.

Table 2 presents the main characteristics of the studied databases and a representation of each image of each database can be found in Appendix A.

#### 3.1.1. Existing Databases

In this section, three databases available online are presented. They all include compression artifacts. Some of them use a backward compatible compression. This method allows the images to be displayable with SDR equipment while preserving the dynamic range for the display on HDR screens. A Tone Mapping Operator (TMO) is used to tone map the HDR images into SDR range. These tone-mapped images are then compressed using different codecs. After the decoding process, they are tone expanded to recover their HDR characteristics. These three databases use the same HDR SIM2 display (HDR47ES4MB) which has a measured dynamic range going from 0.03 to 4250 ${\mathrm{cd}/m}^{2}$.
Narwaria et al. [34]’s database (Available at http://ivc.univ-nantes.fr/en/databases/JPEG_HDR_Images/) is composed of 10 source images, which have been distorted by a backward compatible compression with the JPEG codec and the iCam TMO [36]. This database was used along with others to tune HDR-VDP2. The angular resolution used during the subjective test was 60 pix/degree and the surround luminance was 130 ${\mathrm{cd}/m}^{2}$. For this database due to its surround luminance above 100 ${\mathrm{cd}/m}^{2}$, we consider a surround luminance of 20 ${\mathrm{cd}/m}^{2}$ to obtain the color space $HDR\text{-}Lab_{100}$. The characteristics of each reference image are reported on Figure 3.Korshunov et al. [35]’s database (Available at http://mmspg.epfl.ch/jpegxt-hdr) consists in images distorted with a backward-compatible compression scheme using the JPEG-XT standard and either the Mantiuk et al. [37] or the Reinhard et al. [38] TMO. The angular resolution used during the subjective test was 60 pix/degree and the surround luminance was 20 ${\mathrm{cd}/m}^{2}$. The characteristics of each reference images are reported on Figure 4.Zerman et al. [19]’s database (Available at http://webpages.l2s.centralesupelec.fr/perso/giuseppe.valenzise/) is partially composed of images from [39]. The distorted images are generated by using both backward-compatible compression using the TMO proposed by Mai et al. [40] and using a non backward-compatible compression with the use of the PQ function for the EOTF. The compression was performed using the JPEG and the JPEG2000 codec. The angular resolution used during the subjective test was 40 pix/degree and the surround luminance was 20 ${\mathrm{cd}/m}^{2}$. The characteristics of each reference images are reported on Figure 5.

The main limitation of these databases is the limited BT.709 gamut used during their creation. The wider BT.2020 color gamut is more and more associated with HDR images and videos. In addition, Standards Developing Organizations such as DVB, recommend the use of the BT.2020 gamut with HDR content [41].

#### 3.1.2. Proposed Databases

To deal with the limitations of existing databases, we propose two new databases (Available at www-percept.irisa.fr/software/). The first one is a database with BT.709 contents encapsulated in a BT.2020 gamut and the second one is composed with native BT.2020 content. We used the same display for both of them: the SONY BVM-X300. This is a professional HDR video monitor able to faithfully display the brightness of signals [42]. It has a peak brightness at 1000 ${\mathrm{cd}/m}^{2}$ and a luminance of a black pixel that was too low to be measured by our equipment (<0.2 ${\mathrm{cd}/m}^{2}$). In this paper, we assume a luminance for the black pixel at 0.001 ${\mathrm{cd}/m}^{2}$. This monitor also allows us to force the use of a chosen EOTF without having to consider the image metadata.

To display the images on the screen, we used the b<>com *Ultra Player* which allows to distribute uncompressed YUV content with a 10 bits quantization and 4:2:0 chroma sub-sampling.The connection to the screen was done using 3G-SDI cables.

To create the distortions, we used HDRTools (v0.16) (Available at https://gitlab.com/standards/HDRTools/) to apply format conversion, chrominance sub-sampling or gamut conversion. For the compression and decompression of the images, we used the reference software of HEVC, the HEVC test Model (v16.17) (Available at https://hevc.hhi.fraunhofer.de/).

For both subjective tests, we used the same methodology: the Double-Stimulus Impairment Scale (DSIS) variant I methodology [43] with a side-by-side comparison. Pairs of images were presented to the viewers, one side being always the reference. 50% of the participant had the reference always on the right-hand side, 50% always on the left-hand side. To avoid a bias with the order of presentation, the pairs of images were randomized for each participant with the condition that the same content was never shown twice consecutively. Each image pair was shown 10 s and voting time was 5 s. The test session lasts 35 min (including instructions and training time) with a 5 min pause in the middle of the test.

To obtain realistic distortion we compress the images with the HEVC codec using the recommendation ITU-T H Suppl.15 [44], i.e., a 10 bits quantization for the images, the PQ EOTF and the $Y^{\prime }CrCb$ color space. Moreover this recommendation proposes different processes for increasing the quality of the compression such as a 4:2:0 chroma subsampling using a luma-adjustment process and a chroma Qp adaptation. This last is of special interest for this study because it corrects errors due to the compression of the chrominances. In WCG, most of chroma values tend to be near their mean value (i.e., 512) while the $Y^{\prime }$ component tends to use most of its range. This is even more significant for BT.709 content encapsulated in a BT.2020 gamut. This behaviour creates a shift in the bitrate allocation from the chrominance to the luminance. This can potentially create visible chrominance artifacts. The chroma Qp adaptation proposed in the recommendation is a method to counter this effect. Because we wanted to create realistic color artifacts, we distort the images using the compression with and without the chroma Qp adaptation to study the sensibility of the color metrics.

The first database we propose was already presented in [45]. Eight images were selected from 3 collections: two are from the MPEG HDR sequences (FireEater and Market) [46], one is from the Stuttgart HDR Video Database [47] and the remaining five images are from the HDR photographic survey [48]. Note that these images also belong to Zerman et al.’s database [19]. The characteristics of the images are not exactly the same as in Zerman et al. [19] because we used only half of the images (944×1080) to be able to display simultaneously both the reference image and the distorted image on the same screen. The characteristics of the images can be found on Figure 6.

Fifteen naive subjects participated in this test (11 males, 4 females) with an average age of 25.8. All declared normal or corrected-to-normal vision. One participant was removed from the analysis using the methodology described in [43].

Because we used the display in HD mode, we choose to call this database HDdtb in this paper. Four kinds of distortions have been chosen:HEVC compression using the recommendation ITU-T H Suppl.15 [44]. Four different quantization parameters (Qp) were selected for each image for this distortion.HEVC compression without the chroma Qp adaptation leading to more chrominance distortions. Three Qp were selected for each image.Gaussian noise on the chroma components: 3 levels of noise were selected.Gamut mismatch: two kinds of distortion were created: on one hand, the BT.709 images were considered as if they had been already encapsulated in a BT.2020 gamut leading to more saturated images. On the other hand, we took images already encapsulated in a BT.2020 gamut and considered them as BT.709 images and re-encapsulated them in a BT.2020 gamut. This creates less saturated images.

For the second database, we used eight 4K images produced by Harmonic Inc. The characteristics of the images are given in Figure 7. We used only part of these images so the reference and the distorted image could fit on our display (1890×2160) with a band of 60 black pixels. We called this new database 4Kdtb. We used the same visualization distance as in the previous database. Thirteen experts or sensitized subjects participated in this test (11 males, 2 females) with an average age of 40. All declared normal or corrected-to-normal vision. With this database we aim to create more visible color artifacts induced by compression than in the HDdtb database. We compressed the images with four different Qp with three different options for the compression:HEVC compression using the recommendation ITU-T H Suppl.15 [44].HEVC compression without the chroma Qp offset algorithm.HEVC compression with 8 bits quantization for the chroma instead of 10 during the compression. The chroma were re-sampled to 10 bits before displaying images on screen.

Because we have a higher resolution, the angular resolution increases as well and become 120 pix/degree. Because most quality metrics are not adapted to this kind of resolution, we choose to downsample the images to obtain a 60 pix/degree resolution before testing the different quality metrics. Using a downsampled image can improve the performances of some metrics such as MS-SSIM or HDR-VDP2, which were tuned for lower resolution.

### 3.2. Performance Indexes

We present the performances of the different quality metrics for each database with four different indicators. Before computing these indicators, a non linear regression is applied to the quality scores thanks to a logistic function:(31)$${\tilde{Q}}_{i}=a+\frac{b}{1+e^{-\frac{\left(Q_{i}-c\right)}{d}}}$$
where $Q_{i}$ is the score of the quality metrics on the image *i* and ${\tilde{Q}}_{i}$ the mapped quality score. Parameters *a*, *b*, *c* and *d* are determined by the regression conducted by the lsqcurvefit function of Matlab.

The four performance indicators are given below:the Pearson correlation coefficient (PCC) (cf. Table A1 and Table A2) measures the linearity between two variables:
(32)$${\mathrm{PCC}}_{S,Q}=\frac{\mathrm{cov}(S,Q)}{\sigma_{S}\sigma_{Q}}$$
where, *S* corresponds to the subjective scores (MOS), *Q* to the predicted quality score, cov(S,Q) is the covariance of *S* and *Q* and $\sigma_{S}$ (resp. $\sigma_{Q}$) is the standard deviation of *S* (resp. *Q*).the Spearman rank Order Correlation coefficient (SROCC) (cf. Table A3 and Table A4) measures of the monotony between two variables. Raw scores *S* and *Q* are first converted to ranks $rg_{S}$ and $rg_{Q}$. The SROCC corresponds to the PCC of these two new variables:
(33)$${\mathrm{SROCC}}_{S,Q}=\frac{\mathrm{cov}(rg_{S},rg_{Q})}{\sigma_{rg_{S}}\sigma_{rg_{Q}}}$$the Outlier Ratio [49] (OR) (cf. Table A5 and Table A6) represents the quality metrics consistency. It represents the number of outlier point to total points *N*:
(34)$$\mathrm{OR}=\frac{Total\;of\;Outlier}{N}$$
An outlier is defined as a point for which the error exceeds the 95 percent confidence interval of the mean MOS value as defined in [43].The Root Mean square error (RMSE) (cf. Table A7 and Table A8) measures the accuracy of the quality metrics:
(35)$$\mathrm{RMSE}=\sqrt{\frac{1}{N}\sum_{i=1}^{N}{\left(S\left(i\right)-Q\left(i\right)\right)}^{2}}$$

## 4. Results

In this section, we present the performances of the different metrics presented in the previous sections. For the sake of completeness, we also study the performances of the following color-blind HDR metrics: PU-VIF [9], HDR-VDP2 [11] (version 2.2.1) [12] and HDR-VQM [13]. Note that HDR-VDP2 requires a number of parameters such as the angular resolution, the surround luminance and the spectral emission of the screen. For these parameters, we use the values corresponding to the different subjective tests. We measured the spectrum of the Sony BVM-X300 and the SIM2 HDR47ES4MB monitor using the “X-Rite Eye one Pro 2” probe (more details are given in [45]. All these parameters are summarized in Table 3.

Figure 8, Figure 9, Figure 10, Figure 11 and Figure 12 represent the SROCC performances for each database and each metric. Numerical values of the performance indexes (SROCC, PCC, OR, RMSE) can be found in Appendix B.

### 4.1. 4Kdtb Database

With our proposed 4Kdtb (cf. Figure 8), for each color-blind metrics, the best color spaces are always the ICtCp, $HDR\text{-}Lab_{100}$ and the PU-encoding. $J_{z}a_{z}b_{z}$ and $HDR\text{-}Lab_{1000}$ provide the lowest performances. The best performing color-blind metrics is FSIM used with the PU-encoding, closely followed by ${\mathrm{FSIM}}_{ICtCp}$ and ${\mathrm{FSIM}}_{HDR\text{-}Lab_{100}}$. MS-SSIM used with the PU encoding, ICtCp and, $HDR\text{-}Lab_{100}$ are almost on par with the second best performing metrics HDR-VDP2 (cf. Appendix B). The only color space that provides good performances on all color metrics is the ICtCp color space.

### 4.2. Zerman et al. Database

With the Zerman et al. database, as previously, the color spaces, ICtCp, $HDR\text{-}Lab_{100}$ and the PU-encoding provide the best performances for almost all color-blind metrics (cf. Figure 9). However, there is one exception with FSIM. Used with the following color spaces, $J_{z}a_{z}b_{z}$, $HDR\text{-}Lab_{100}$ and $HDR\text{-}Lab_{1000}$, it provides slightly better performances than ICtCp and the PU-encoding. The best performing color-blind metrics are, with almost the same performances, HDR-VDP2, HDR-VQM and ${\mathrm{MS}\text{-}\mathrm{SSIM}}_{ICtCp}$.

### 4.3. HDdtb Database

With our proposed HDdtb (cf. Figure 10), for color-blind metrics, the color space $J_{z}a_{z}b_{z}$ provides slightly lower performances for all metrics, except with FSIM. For this metric, the performances with $J_{z}a_{z}b_{z}$ are higher. The best performing color-blind metrics for this database are ${\mathrm{FSIM}}_{J_{z}a_{z}b_{z}}$, ${\mathrm{FSIM}}_{HDR\text{-}Lab_{1000}}$ and ${\mathrm{MS}\text{-}\mathrm{SSIM}}_{HDR\text{-}Lab_{100}}$. For the color metrics, the metrics based on color difference metrics ($\bar{\Delta E}$, $\bar{\Delta E^{S}}$ and CSSIM) have very low performances. This is partially due to the presence of the gamut mismatch artifact. As noticeable on Table 4, discarding this artifact increases the performances of these metrics. For the participants of our subjective test, the distortions on the images are clearly visible but are not directly associated with a loss in quality perception.

### 4.4. Korshunov et al. Database

The Korshunov et al. database is the less selective database (cf. Figure 11). Most of the metrics have high correlation coefficients and the choice of color space has close to no impact on the performances especially on color-blind metrics. Even using non-perceptually linear color space like the $Y^{\prime }Cr^{\prime }Cb^{\prime }$ color space impacts only moderately the performances of MS-SSIM, FSIM, PSNR-HVS-M and PSNR-HMA. For this database, the best performing color-blind metrics are ${\mathrm{FSIM}}_{J_{z}a_{z}b_{z}}$, ${\mathrm{FSIM}}_{HDR\text{-}Lab_{1000}}$ and ${\mathrm{MS}\text{-}\mathrm{SSIM}}_{J_{z}a_{z}b_{z}}$.

### 4.5. Narwaria et al. Database

With the Narwaria et al. database (cf. Figure 12), $J_{z}a_{z}b_{z}$ is the best color space for SSIM and MS-SSIM while the PU-encoding and the $HDR\text{-}Lab_{100}$ are the best color spaces for FSIM. The best metrics for this database are ${\mathrm{MS}\text{-}\mathrm{SSIM}}_{J_{z}a_{z}b_{z}}$, HDR-VDP2 and HDR-VQM. The good performances of HDR-VDP2 were expected for this database because it was part of the training set of this metric. For this database, the performances of the PSNR and the PSNR-HVS-M are relatively low compared to the other databases. The fact that PSNR-HMA with the adequate color space significantly increases the performances of PSNR-HVS-M suggests that the backward compatible compression used by Narwaria et al. (Section 3.1.1) creates distortions that impact the mean luminance and the contrast of the images. Indeed PSNR-HMA is an improvement of PSNR-HVS-M that takes into account these two kinds of artifacts [50].

### 4.6. Results Summary

For all studied databases, HDR-VDP2 has generally high performances although it is not always on the top three metrics (cf. Appendix B). FSIM and MS-SSIM with appropriate perceptually uniform color space are often on par if not better than HDR-VDP2.

Among all metrics, FSIM is the less sensitive metrics to the choice of color space assuming that this color space is perceptually uniform.

The color extension of FSIM, namely FSIMc, does not improve the performances of FSIM even for our proposed database 4Kdtb which focuses on chromatic distortions. Worst, the metrics becomes much more sensitive to the color space choice. We observe the same behavior for the color extension of PSNR-HMA, PSNR-HMAc, which decreases the performances of the metrics for any color spaces.

When using the two non-uniform color space XYZ and $Y^{\prime }Cr^{\prime }Cb^{\prime }$, the performances of all metrics drop significantly compared to the other color spaces for all the databases and especially for our proposed database 4Kdtb, the Zerman et al. database and the Narwaria et al. database. It emphasizes the importance of perceptually uniform color space for predicting the quality of HDR images.

## 5. General Discussion

We separate our general discussion in two parts. First, we study the impact of the color space on the metrics performances. Moreover we emphasize the influence of the diffuse white luminance. As a reminder, the luminance of the diffuse white correponds to the luminance of a 100% reflectance white card without any specular highlight. In HDR imaging, it is different from the peak brightness. In the second part of our analysis, we discuss the sensitivity of chrominance artifacts on color metrics using our proposed database 4Kdtb.

### 5.1. Impact of the Diffuse White Luminance

Our results suggest that the best color space for assessing the quality of HDR images depends on the test database. Indeed, some of the color spaces are adapted and tuned for one visualization condition.

The HDR-Lab color space considers two important parameters, i.e., the diffuse white and the surround luminance. Moreover, the final equation of the $J_{z}a_{z}b_{z}$ color space (Equation (13)) was tuned using the experimental dataset called SL2 [20]. This dataset was obtained for a diffuse white at 997 ${\mathrm{cd}/m}^{2}$. This explain why the $J_{z}a_{z}b_{z}$ luminance have a behaviour close to the $HDR\text{-}Lab_{1000}$ luminance (cf. Figure 2).

The PU function and the ICtCp color space were not obtained through the same kind of training. They were created using Daly’s Contrast Sensitivity Function model [51] and Barten’s Contrast Sensitivity Function model [52], respectively. However, Figure 2, that represent the different color spaces luminance in function of the linear luminance, suggest that ICtCp and the PU encoded luminance have a behaviour closer to the $HDR\text{-}Lab_{100}$ luminance than from the $J_{z}a_{z}b_{z}$ luminance or than the $HDR\text{-}Lab_{1000}$ luminance.

Because the color spaces are adapted for different viewing conditions, it is not easy to determine the best color space.
With the proposed database 4Kdtb, the color spaces with a diffuse white around 100 ${\mathrm{cd}/m}^{2}$ (ICtCp, $HDR\text{-}Lab_{100}$ and the PU-mapping) give better performances than $J_{z}a_{z}b_{z}$ and $HDR\text{-}Lab_{1000}$ spaces. We also observe that the performances of color metrics are more sensitive to the color space choice.We draw a similar conclusion on Zerman et al. database, except for FSIM and FSIMc (cf. Figure 9). These two metrics are less sensitive to the color space for this database.With the proposed database HDdtb (cf. Figure 11), the $J_{z}a_{z}b_{z}$ color space provides the lowest performances for PSNR, SSIM, MS-SSIM and PSNR-HMA metrics but provides the highest performances with FSIM and FSIMc. However, results indicate that the PSNR, SSIM and PSNR-HMA metrics based on $HDR\text{-}Lab_{1000}$ and $HDR\text{-}Lab_{100}$ color spaces perform better than the same metrics using the $J_{z}a_{z}b_{z}$ color space. This suggests that the low performances of these metrics is not due to the diffuse white characteristics of the images, but to the design of $J_{z}a_{z}b_{z}$ color space which corrects a deviation in the perception of the blue hue (cf. Equation (9)). To test this hypothesis, we measure the SROCC of these metrics on the HDdtb database with the $J_{z}a_{z}b_{z}$ color space without the blue deviation correction. We call this new space $\tilde{J_{z}a_{z}b_{z}}$. Results, shown in Table 5, indicate that SROCC values of the three aforementioned metrics increase with the $\tilde{J_{z}a_{z}b_{z}}$ color space. In addition, metrics using this modified color space provide similar performances to metrics based on the $HDR\text{-}Lab_{1000}$ color space. This is consistent with the fact that $HDR\text{-}Lab_{1000}$ and $J_{z}a_{z}b_{z}$ are adapted to almost the same diffuse white luminance. This might be due to the presence of the “gammut mismatch” artifact in this database. Indeed, the “gammut mismatch” artifact creates visible distortions that was not associated with a subjective quality loss during our test. We suspect that the blue hue deviation correction makes the $J_{z}a_{z}b_{z}$ color space more sensitive to this distortion. However, this is difficult to demonstrate due to the low number of images with this kind of artifact present in this database.With the Narwaria et al. database, it is difficult to draw a conclusion (cf. Figure 12). The MS-SSIM and the SSIM metrics perform better when using the $J_{z}a_{z}b_{z}$ color space. However, the FSIM and PSNR-HMA metrics perform better when using the ICtCp color space. This contrasted result might be due to the fact that the diffuse white luminance is likely not homogeneous across the entire database.

To go further into the analysis, we propose to evaluate the impact of the diffuse white on the performances of HDR-Lab metrics. The SROCC performances of three metrics (FSIM, MS-SSIM and SSIM) are evaluated for a diffuse white in the range 80 to 1000. Results are plotted in Figure 13. For the FSIM, the performances decrease slightly when the diffuse white luminance increases for the 4Kdtb database and the Narwaria et al. database while increasing with the diffuse white for the HDdtb. The impact of the diffuse white is more important on the MS-SSIM metric. For example, with the Zerman et al. database, the SROCC score drops from 0.9143 to 0.7791. The impact for the SSIM metrics is in the same order of magnitude as for MS-SSIM.

### 5.2. Sensibility to Chrominance Distortions

In this section, the ability of color metrics to take into account chrominance artifacts is discussed. The discussion is focused on the database 4Kdtb which is the only database providing significant chrominance artifacts. Also we only consider metrics using the ICtCp color space since the best performances are observed with this color space. Figure 14 presents the Mean Opinion Score (MOS) and objective scores for the reference image “Regatta_24s”, for the distorted images (compressed with HEVC). The objective scores are given after applying the logistic regression presented in Section 4. Results for the other reference images can be found on Appendix C.

There is a clear difference of quality perception between the images compressed with the chroma Qp adaptation (cf. Section 3.1.2) (red Line) and the images compressed without the chroma Qp adaptation and a 8 bits quantization on the chrominance (blue line). The MOS of images compressed without the chroma Qp adaptation algorithm and a 10 bits quantization (green line) are in-between the two previous encodings.

As expected, the color-blind metrics, i.e., HDR-VDP2 and FSIM, are not sensitive at all to the chrominance distortions. However, more surprisingly, the color extension of FSIM, namely FSIMc, is not sensitive to the generated chrominance artifacts. The metrics was tailored for images in a BT.709 gamut with a SDR range. Its non-sensibility to the chrominance might be due to the pre-defined constant used for the color comparisons [29].

The other color metrics, i.e., $\bar{\Delta E^{S}}$, SSIMc and CSSIM, are more sensitive to the chrominance artifacts. However, SSIMc and CSSIM have a tendency to underestimate the influence of chrominance artifacts for images compressed with a low Qp (so low distortion in the luminance channel) and a 8 bits quantization on the chrominance (cf. Figure A7, Figure A9, Figure A10, Figure A11 and Figure A13).

## 6. Recommendations

In this section, some recommendations are given to assess the HDR/WCG content quality in the context of image/video compression. The recommendations are listed below:For assessing the impact of luminance distortions, we recommend to use the FSIM metric. This is one of the best performing metrics. Moreover, it is the less sensitive to the choice of color space and to the diffuse white of the images. Using the color extension of the metrics (FSIMc) does not bring a significant added-value. In addition, it is important to underline that the FSIMc metrics is sensitive to the choice of color space (cf. Figure 8).For assessing the impact of chrominance distortions, we recommend to use the ${\bar{\Delta E^{S}}}_{ICtCp}$ metric.For assessing the impact of both luminance and chrominance distortions, we recommend to use both the FSIM metrics and the ${\bar{\Delta E^{S}}}_{ICtCp}$ metrics.To choose the color space, we recommend to take into account the diffuse white used during the color grading of the images. If the producer of the content follows the ITU recommendation BT.2408 [53] that defines the diffuse white luminance at 203 ${\mathrm{cd}/m}^{2}$, we recommend to use the ICtCp color space. Indeed, this color space is well adapted to a low value of diffuse white. At the opposite, the $J_{z}a_{z}b_{z}$ color space is well appropriate for a diffuse white luminance at 997 ${\mathrm{cd}/m}^{2}$. Another benefit to use the ICtCp color space is related to its direct compatibility with popular compression codecs such as HEVC.For application where the calculation time and the complexity are critical aspects, we recommend to be very careful with the choice of the color space. The simplest metrics, such as PSNR and SSIM, are much more sensitive than FSIM to the diffuse white luminance.If the chosen metrics is the PSNR, we recommend to first verify that the tested image/video processing application, such as compression codecs, does not create luminance mean shift or contrast change. These artifacts can be induced by backward compatible compression (if the image is first tone-mapped, then compressed using a legacy codec and finally tone expanded).

Due to the characteristics of the tested databases, these recommendations have to be used in the context of image/video compression. Different subjective tests would be required to extend the analysis to other kinds of distortion.

## 7. Conclusions

In this article, we reviewed the relevance of using SDR metrics with perceptually uniform color spaces to assess the quality of HDR/WCG contents. We studied twelve different metrics along with six different color spaces. To evaluate the performances of these metrics, we used three existing HDR image databases annotated with MOS and created two more databases specifically dedicated to WCG and chrominance artifacts. We showed that the use of perceptually uniform color spaces increases, in most cases, the performances of SDR metrics for HDR/WCG contents.

In this study, we also highlight two weaknesses of state-of-art metrics. First, The relationship between the diffuse white used for grading the image and the diffuse white used for the color space is not always easy to define. In a number of cases, we do not know the value of the diffuse white used for grading of the image. Choosing an arbitrary diffuse white for the color space may significantly alter the objective quality assessment. Further analysis of this relationship is required. A better understanding could help to evaluate compression of images using the HLG EOTF for which the diffuse white depends on the display. Second, to the best of our knowledge, the quality assessment of DR/WCG images with chrominance distortions is still an open-issue, because of the lack of relevant objective metrics.

In a broader perspective, the relevance of subjective tests can also be questioned. For example, on the proposed database HDdtb, viewers did not perceive the gamut mismatch artifact as a loss of quality. However, this kind of artifact changes completely the appearance of images. Some other artifact could also alter the image appearance like the tone mapping/tone expansion used during backward compatible compression. In some cases, asking the viewers to not assess only the quality of the images but also their fidelity to the image appearance can be valuable to fully evaluate image processing algorithms.

## Figures and Tables

**Figure 1 jimaging-05-00018-f001:**
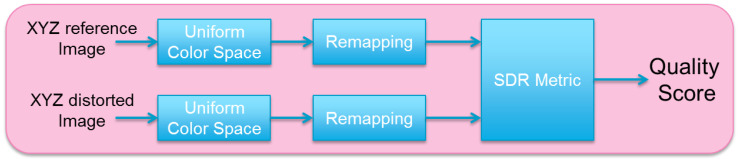
Diagram of the proposed method to adapt SDR metrics to HDR/WCG contents.

**Figure 2 jimaging-05-00018-f002:**
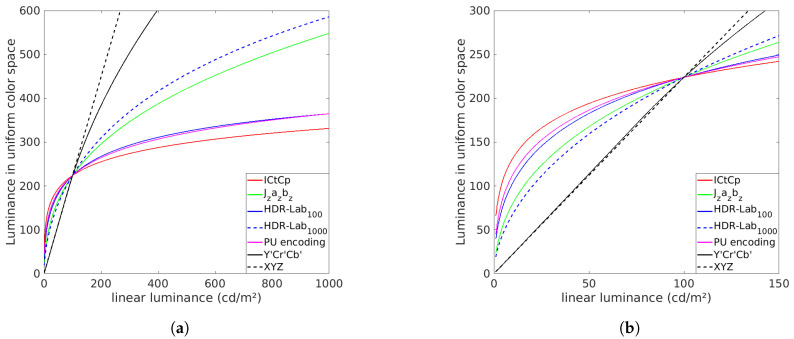
Different perceptually uniform luminances as a function of the linear luminance: (**a**) for the range 0–1000 ${\mathrm{cd}/m}^{2}$, (**b**) for the range 0–150 ${\mathrm{cd}/m}^{2}$.

**Figure 3 jimaging-05-00018-f003:**
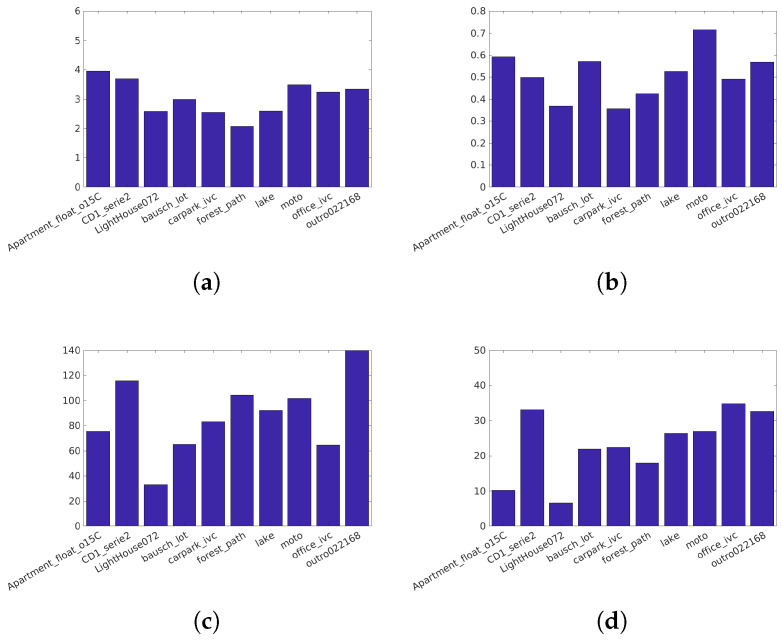
Characteristics of the Narwaria et al. [34] images: (**a**) The dynamic range, (**b**) key, (**c**) spatial Information, (**d**) Colorfulness.

**Figure 4 jimaging-05-00018-f004:**
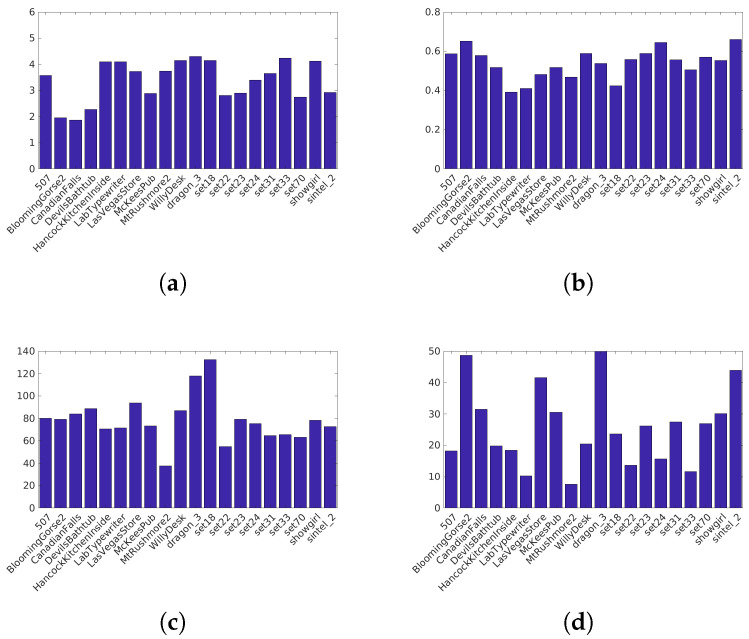
Characteristics of the Korshunov et al. [35] images: (**a**) The dynamic range, (**b**) key, (**c**) spatial Information, (**d**) Colorfulness.

**Figure 5 jimaging-05-00018-f005:**
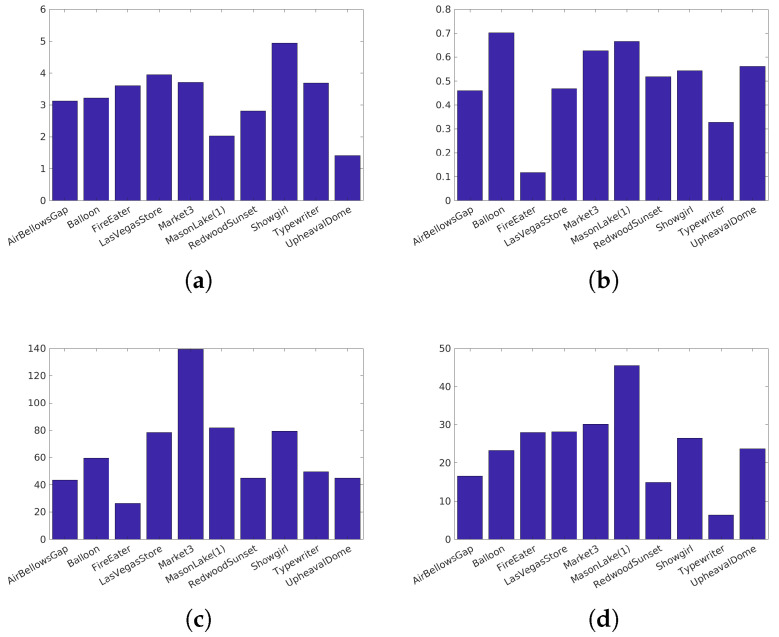
Characteristics of the Zerman et al. [19] images: (**a**) The dynamic range, (**b**) key, (**c**) spatial Information, (**d**) Colorfulness.

**Figure 6 jimaging-05-00018-f006:**
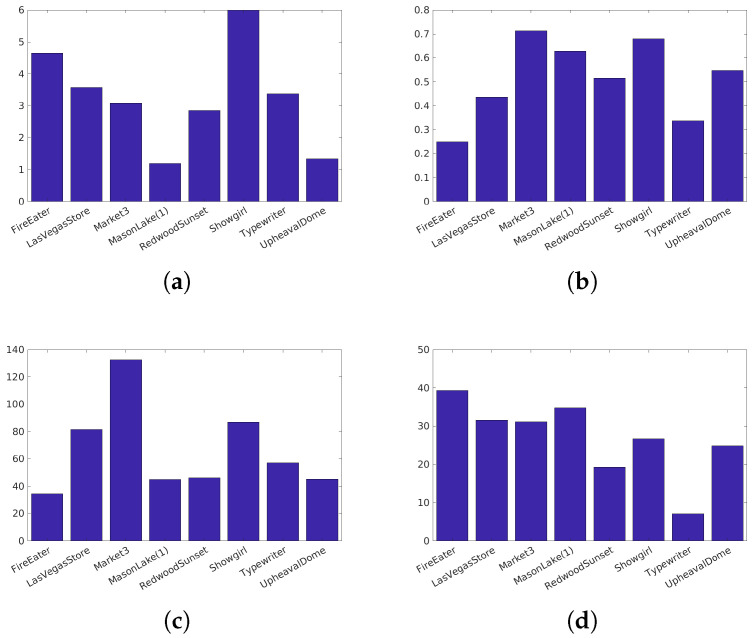
Characteristics of the HDdtb images: (**a**) The dynamic range, (**b**) key, (**c**) spatial Information, (**d**) Colorfulness.

**Figure 7 jimaging-05-00018-f007:**
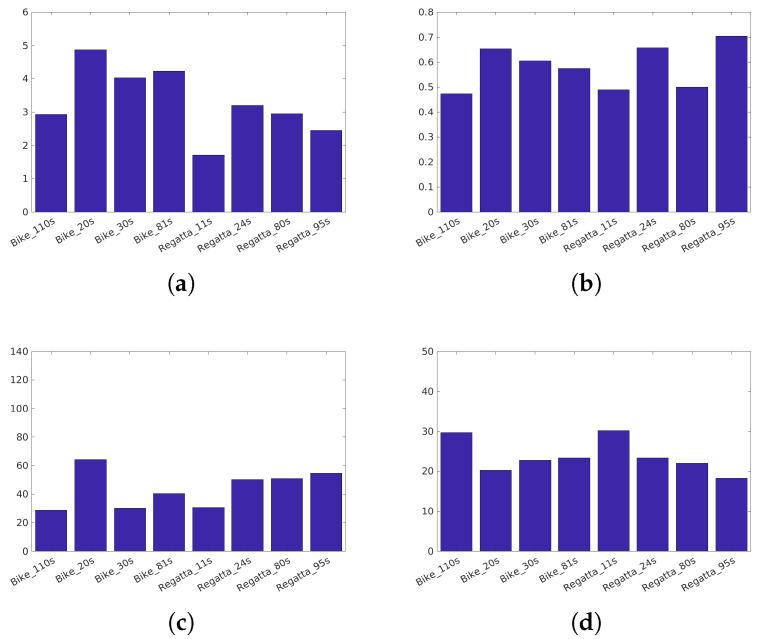
Characteristics of the 4Kdtb images: ((**a**) The dynamic range, (**b**) key, (**c**) spatial Information, (**d**) Colorfulness.

**Figure 8 jimaging-05-00018-f008:**
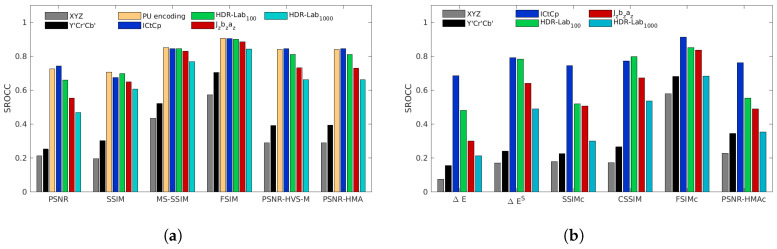
SROCC performances for the 4Kdtb database for color-blind quality metrics (**a**) and for color quality metrics (**b**).

**Figure 9 jimaging-05-00018-f009:**
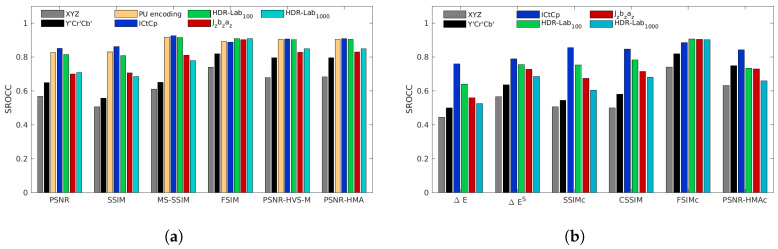
SROCC performances for the Zerman et al. database for color-blind quality metrics (**a**) and for color quality metrics (**b**).

**Figure 10 jimaging-05-00018-f010:**
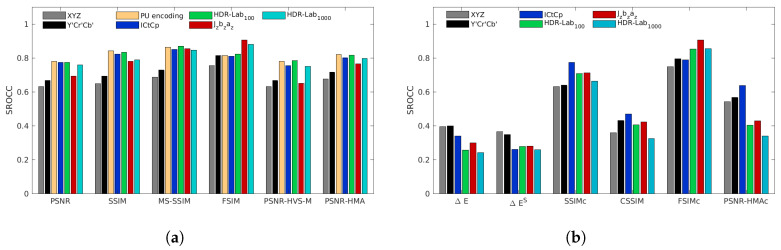
SROCC performances for the HDdtb database for color-blind quality metrics (**a**) and for color quality metrics (**b**).

**Figure 11 jimaging-05-00018-f011:**
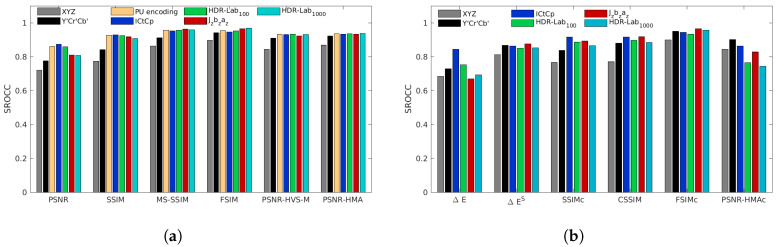
SROCC performances for the Korshunov et al. database for color-blind quality metrics (**a**) and for color quality metrics (**b**).

**Figure 12 jimaging-05-00018-f012:**
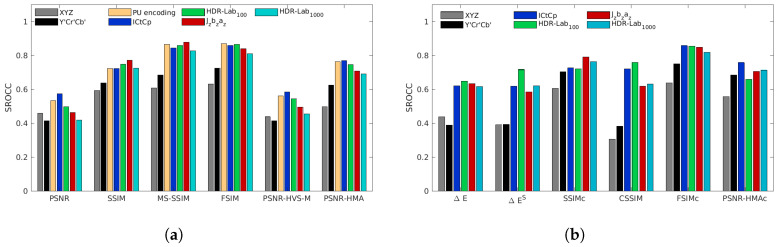
SROCC performances for the Narwaria et al. database for color-blind quality metrics (**a**) and for color quality metrics (**b**).

**Figure 13 jimaging-05-00018-f013:**
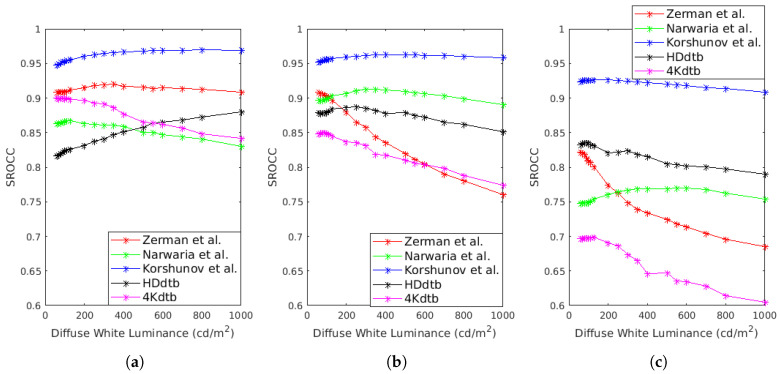
SROCC of (**a**) ${\mathrm{FSIM}}_{HDR\text{-}Lab}$, (**b**) ${\mathrm{MS}S\text{-}\mathrm{SSIM}}_{HDR\text{-}Lab}$, (**c**) ${\mathrm{SSIM}}_{HDR\text{-}Lab}$ in function of the diffuse white luminance.

**Figure 14 jimaging-05-00018-f014:**
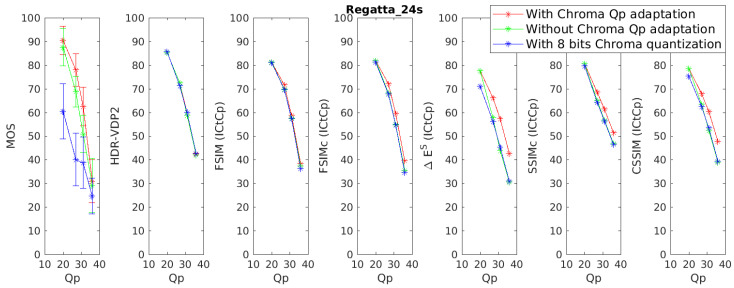
Subjective and objective scores for the image Regatta_24s and for 6 metrics based on the ICtCp color space.

**Table 1 jimaging-05-00018-t001:** Selected SDR quality metrics.

Name	Color	Reference	Main Principle
**PSNR**			Ratio between the range of the signal and the mean square error
$\bar{\Delta E}$	✓		Mean of the color difference metrics
$\bar{\Delta E^{S}}$	✓	Zhang et al. [26]	Mean of the color difference metrics by considering the blurring effect of the HVS. Also known as S-CIELab
**SSIM**		Wang et al. [8]	Metrics based on the comparison of three characteristics of the images: the luminance, the contrast and the structure
**SSIMc**	✓	Wang et al. [27]	Linear combination of the SSIM applied on the three components Y${}^{\prime }$, Cr and Cb of the images.
**CSSIM**	✓	Hassan et al. [28]	Combination of SSIM and $\Delta E^{S}$
**MS-SSIM**		Wang et al. [10]	Multi-scale SSIM
**FSIM**		Zhang et al. [29]	Comparison of the phase congruency and the gradient magnitude
**FSIMc**	✓	Zhang et al. [29]	Color extension of FSIM. Adds two comparisons corresponding to the two chrominance components
**PSNR-HVS-M**		Ponomarenko et al. [30]	PSNR on the DCT blocks of the images using CSF and visual masking
**PSNR-HMA**		Ponomarenko et al. [31]	Improvement of the PSNR-HVS-M. Takes into account the particularities of the mean shift and the contrast change distortions
**PSNR-HMAc**	✓	Ponomarenko et al. [31]	Linear combination of the PSNR-HMA on the three components Y${}^{\prime }$, Cr and Cb of the images.

**Table 2 jimaging-05-00018-t002:** Number of observers, number of images, subjective test protocol, kind of distortion and used display for the 3 existing HDR image quality databases and the two databases proposed in this paper.

Name	#Obs	#Img	Protocol	Distortion	Display	Gamut	Size
Narwaria et al. [34]	27	140	ACR-HR	JPEG	SIM2 HDR47ES4MB	BT.709	1920 × 1080
Korshunov et al. [35]	24	240	DSIS (side by side)	JPEG-XT	SIM2 HDR47ES4MB	BT.709	944 × 1080
Zerman et al. [19]	15	100	DSIS	JPEG, JPEG-XT JPEG2000	SIM2 HDR47ES4MB	BT.709	1920 × 1080
Proposed HDdtb	15	96	DSIS (side by side)	HEVC, Gaussian noise, Gamut mismatch	Sony BVM-X300	BT.2020	944 × 1080
Proposed 4Kdtb	13	96	DSIS (side by side)	HEVC, Quantization	Sony BVM-X300	BT.2020	1890 × 2160

**Table 3 jimaging-05-00018-t003:** Parameters used for HDR-VDP2.

Name	Angular Resolution (Pixel/Degree)	Surround Luminance (${\mathrm{cd}/m}^{2}$)	Spectral Emission
Narwaria et al.	60	130	SIM2 HDR47ES4MB
Korshunov et al.	60	20	SIM2 HDR47ES4MB
Zerman et al.	40	20	SIM2 HDR47ES4MB
HDdtb	60	40	Sony BVM-X300
4Kdtb	60	40	Sony BVM-X300

**Table 4 jimaging-05-00018-t004:** SROCC for the HDdtb database with and without the gamut mismatch artifact.

Quality Metric	All Images	Without the “Gamut Mismatch” Distortion	Compression Artifacts Only
${\bar{\Delta E}}_{HDR\text{-}Lab_{100}}$	0.2578	0.3905	0.6190
${\bar{\Delta E^{S}}}_{HDR\text{-}Lab_{100}}$	0.2784	0.5687	0.6946
${\mathrm{CSSIM}}_{HDR\text{-}Lab_{100}}$	0.4065	0.6453	0.7714

**Table 5 jimaging-05-00018-t005:** SROCC for the HDdtb database for three metrics based on $J_{z}a_{z}b_{z}$, $\tilde{J_{z}a_{z}b_{z}}$ and $HDR\text{-}Lab_{1000}$.

	Color Spaces		
Metrics	$J_{z}a_{z}b_{z}$	$\tilde{J_{z}a_{z}b_{z}}$	$\mathrm{HDR}-{\mathrm{Lab}}_{1000}$
PSNR	0.6933	0.7463	0.7587
SSIM	0.7831	0.7973	0.7904
PSNR-HMA	0.7664	0.7949	0.7984

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
