# Peer review of "Quality Assessment of HDR/WCG Images Using HDR Uniform Color Spaces"

_2313-433X, 2019, doi:10.3390/jimaging5010018_

Round 1

Reviewer 1 Report

The authors study in this work multiple objective metrics for WCG and HDR imaging on their own, and several third party databases. In particular, one of the concerns of the authors is the inclusion of color in the study, which is certainly missing. Most if not all image quality indicies suitable for HDR are color blind, or are either only concerned with color impression but not image quality, such as \Delta E. In so far, the study provided by the authors is helpful.

My main concern with this work is, however, that it is quite unbalanced and there is no significant novelty in the work. It is unbalanced because about half of the work reviews exisiting image quality indices probably known to the reader. This can be shortened a lot, and the authors can simply quote the references without going as much into tdetail. It is not novel because the only contribution from the authors is the inclusion of a color transformation and a bit of tweaking of the quality indices.

The problem with this approach is that it misses important details on the human visual system; in particular, the contrast sensitivity function for chrominance differences is quite different from the CSF for luminance perception: The former has a low-pass characteristics, whereas the latter is a bandpass characteristics. It is thus not at all clear why a luminance based quality index can be carried over to color without any further modification of any (explicit or implicit) CSF weight. Same goes for visual masking - masking in the luma domain does not simply carry over to chroma.

Several details are missing from the work: In the comparison, how was HDR-VDP2 calibrated/configured? This can be of major importance for its performance and information on this need to be provided.

Other details:

In section 2.1: Calling the X and Z coordinates of the XYZ color space "chrominance information" is at least highly misleading as it may imply that they encode some sort of color difference. This is not true, however. X,Y and Z are always non-negative, and defined such that every observable color has three non-negative coordinates with Y encoding luminance. XYZ is defined by the CIE standard observer curve. Also, color spaces are not quite as equivalent as the authors may seem to imply. CIELab/Jab and related are derived from XYZ space and are mostly concerned for metrological applications, whereas ICtCp is a color-decorrelation transformation for compression purposes that operates in non-linear coordinate space (gamma-corrected in SDR space, potentially PQ-mapped or PU-mapped for HDR). It is a derived color space.

2.1.2: As stated above, I would probably strike all these details here, but if the authors indent to present them, I would prefer a unified presentation with either all fractions or decimals. What would be more helpful is probably give an idea what I is L+M, Ct = L+S-M and Cp=L-M (approximately) which comes from some models of the HVS.

2.2: Same here: The authors present here formula, but either the presentation is superfluous because it can be found in the literature (so leave it away) or it is too short to motivate them appropriately, e.g. why is it a good idea to model \Delta H_z with a sin? I would suggest to shorten section 2 radically.

2.3.2: Minor: Bad typesetting: Please use \exp and \log in LaTex to set the functions, otherwise they come out in italic, which is not appropriate.

Section 3: What is the "capacity" of a display? I am not aware of this term, it might be helpful to explain it quickly.

Section 3.1.2: Note that you use a monitor different from other works, especially with a lower peak luminance. I assume that the SONY BVM-X300 is a consumer-style monitor for HDR-TV? This may make results less comparable as such monitors tend to dim the backlight in the presence of a light scene in order to also reduce power consumption and heat production.

Section 3.2: Performance indices: I would recommend to state earlier that you do use fitting (which is a standard procedure and recommended) - the way how it is currently presented is confusing as reading (78) seems to imply that PCC is computed on the unfitted (raw) data.

Section 4: Note again (as stated above) that is important to deliver calibration information for HDR-VDP2 as results may be quite different.

Section 4: Did you attempt to measure the statistical significance of the quality indices you looked into? Is there a significant difference in their performance? Does the significance depend on the database?

Section 4, p. 23, line 391: YZ is certainly not a non-linear colourspace. YCbCr is (typically) as it operates on (gamma corrected/PQ-mapped/PU-mapped) coordinates, see notes above. Do you probably mean X'Y'Z' with ' indicating the nonlinear mapping?

Figure 22: I wonder why only the leftmost figure has error bars. If it has, is this observable difference in correlation with or without QP adaption statistically significant or just an artifact? One may also wonder why QP-adaption is the right tool to study the color-significance of quality indices. It is certainly one known effect due to the (probably naive) MPEG method of how to handle chroma, though other color-encoding methods exist (such as in JPEG XT). What about your databases with wrong-gammut encoding, or 4:4:4 vs. 4:2:0 chroma subsampling?

Author Response

Dear reviewers,

Please find our answers to your remarks in the link Word document

Best Regards,

Maxime Rousselot

Reviewer 2 Report

Please look the submitted document draft for edits and suggestions below, divided in two categories:

Scientific / technical questions (green highlight): please consider answering the noted questions as this clarifies the experimental methodology and, ultimately, improve scientific soundness. For example, in a few places it is either unclear which color bit-depth is used, whether it is checked to be consisten throughout the pipeline (from still-image codevalues through viewing application, GPU, video cablings and, finally, displays). HDR imaging can only be processed with a color-depth of, at least, 10 bits/channel (i.e. 30 bits/pixel), if not even higher. As soon as one component of the pipeline (from the above listed) is using 8 bits/channel depth, the HDR imaging just becomes SDR embedded in HDR because the viewing device might go into SDR mode. So the authors should take great care in:

describing the graphis setting for all experiments (hardware and software equipment used, particularly as relates to Operating System, software applications to view/process/save the content, GPU, video cables' type and connectivity, intermediate devices --if any--, display devices);

making sure the color-depth is wide enought in all the experiments in HDR/WCG modes (including those where BT.709 gamut is embedded within BT.2020 gamut);

eventually repeating the tests (or changing their setting) for those with HDR/WCG modes with lower colour-depths.

English language and style/design minor issues (yellow highlight, or redlined): please correct the few English-language typos and the less-than-few mathematical symbols' typos (mostly multiplication dots and missed subscriptings). Make great care at applying a consistend policy for italicized text. No matter what policy authors chose, as long as it's consistent thoughout the paper

Author Response

Dear reviewer,

Please find our answers in the link woed file.

Regards

Maxime Rousselot

Reviewer 3 Report

This paper presents a thorough analysis of several standard dynamic range (SDR) image quality metrics using perceptually uniform color spaces and benchmarks them using five different high dynamic range (HDR) and wide color gamut (WCG) image quality datasets. After this broad analysis, useful insights were provided for HDR/WCG image quality assessment.

The work presented in this paper seems novel and relevant. Except for a few minor issues, the scientific methods used seem sound and the work is presented well. I present my comments below:

Regarding the essence of the paper, I have three minor comments and requests, two of which is on the scientific method used and the other one is on the presentation:

 * In subsection 2.4, I believe the selection of 100 cd/m2 as a normalization point for all the other color spaces should be better explained. I presume that the readers will be confused and will think that this selection would bias the results. 

 * In subsection 3.1.2, it is mentioned the images in 4Kdtb are downsampled. I believe the downsampling factor should be explicitly stated and it should be thoroughly justified, as it can be also seen as an alteration of the images from their original state.

 * Regarding the presentation, I noticed that the text parts of subsections '3.1.1. Existing databases' and the first part of the '3.1.2. Proposed databases' are very similar to [42]. Even though authors of this paper and [42] are the same, I believe that the text should be somewhat modified.

 Apart from these points mentioned above, I have some other minor comments regarding the presentation and the language of the paper:

  - Perceptually Uniform (PU) encoding was renamed as "Perceptual Unit (PU)" in the paper. In the original publication [7], the former term used, and the "Perceptually Uniform (PU) encoding" term should be also used in this paper. Please correct this usage in:

     + line marked 45 on the second page

     + line marked 529 on the 28th page (abbreviations)

 - Comments related to the equations:

     + In its original publication [25], the Eqn. 21 (i.e. Delta_ICtCp) has a 720x multiplier. Please add a 720x multiplier to this equation or clarify that you use a modified version.

     + Eqn. 54 on the 10th page seems to be missing "V(D)" term in the denominator. Please also update the text defining "V(.)" function.

     + Eqn. 55 seems to be missing a "+ E_norm/C[m,n]" part ın the 'otherwise' case.

     + Instead of using \cdot and * operators, please select one single operator (either \cdot or \times) to denote your scalar multiplication operations. E.g.:

         = Eqn. 5 on the third page

         = Eqns. 41,42,47,48 on the ninth page

         = Eqn. 67 on the 11th page

     + Please correct subscripts on Eqn. 15.

     + Please check Eqns. 41,42,47,48 for any other minor notation errors.

 - I believe that the text in the following parts needs to be updated:

     + lines 79-82 on the third page. Please update the structure by mentioning that the color difference metrics were described in the second part and the quality metrics were described in the third part.

     + line marked 83 on the third page (i.e. subsection 2.1. title): I believe it should be "Perceptually uniform color spaces"

     + line marked 106 on the third page: Please rewrite this sentence.

     + Please check Eqn. refs on lines marked 152 and 162 (i.e. equation 29 and 36). I believe that they need to be swapped.

     + line marked 207 on the 12th page: "is the value in the PU space of the luminance when its value is 100 cd/m2 in the linear space." --> "is the luma value in the PU space when linear luminance value is 100 cd/m2."

     + I believe that the following 'luminance' terms should be changed with 'luma':

         = Figs. 1 and 2

         = lines marked 410,411 on the 23rd page

     + Please check and correct the colored highlights in Tables A1-A4. As far as I noticed, some of the metrics were miscolored, especially in the HDdtb and Korshunov et al. columns.

     + Related to the previous point, please update the text in the 22nd page accordingly.

     + Please rewrite the first paragraph of subsection '5.1. Impact of the diffuse white luminance', as this paragraph has many typos.

     + Please check line marked 499 on the 26th page. I believe there are 'five' different color spaces together with HDR-Lab, ICtCp, Jzazbz, XYZ, and Y'Cb'Cr'.

     + I believe the name of the [Ref 42] was misspelled (i.e. line marked 644 on the last page). 

 - There are also some typos which need to be fixed:

     + line marked 38 on the second page: "really closed." --> "really close."

     + line marked 41 on the second page: "are a key tool" --> "are key tools"

     + line marked 44 on the second page: "higher range of dynamic" --> "higher dynamic range"

     + line marked 81 on the third page: "Finally, present" --> "Finally, we present"

     + line marked 84 on the third page: "image processings" --> "image processing applications"

     + line marked 123 on the fifth page: "are a good tool" --> "are good tools"

     + line marked 191 on the 11th page: "C2=0.025" --> "C2=0.25"

     + line marked 206 on the 12th page: "similar range than" --> "similar range as"

     + line marked 373 on the 22nd page: "Pu-encoding ant" --> "PU-encoding and"

     + line marked 410 on the 23th page: "have a closer behaviour of" --> "have a behavior closer to"

     + line marked 459 on the 25th page: "compression, difference mostly" --> "compression, which is mostly"

     + line marked 478 on the 26th page: "its performances" --> "its performance"

     + line marked 503 on the 27th page: "we also highlights" --> "we also highlight"

Author Response

Dear Reviewer,

Please, find our answers to your remarks in the link Word file.

Best regards,

Maxime Rousselot

Round 2

Reviewer 1 Report

As a reviewer, I want to thank the authors for the quick update of their work to have it available for review so readily. While some of the concerns were addressed, I am still concerned with the overall novelty of the paper and the amount of contribution of the authors relative to the size of the paper. A lot of known and trivial notation is introduced that can and should just go. It is not necessary to define PCC, SROOC, OR, RMSE, or the known color spaces ICtCp or Jab or \Delta E as they can be replaced by references to the literature. The whole section 2 should be cut down tremendously, and can be replaced by a table with a list of color spaces and their references. In most cases, details are not relevant for this work, and only where they are, the authors are invited to become a bit more explicit. On page 16, you are still not explicit on the HDR-VDP calibration. Note that this requires knowledge of the test parameters used for the databases, so I really suggest to create a table and list them explicitly, well noting that this cannot always be done and some parameters need to be guessed. You may also want to contact the corresponding authors to receive missing details. On page 19, please fix the missing references ("??" instead of a figure number). On the same page, you seem to suggest that the blue-correction of Jab is responsible for the performance of MSSIN relative to FSIM. However, this should be trivial to asses by simply removing the correction and re-run the numerical evaluation - otherwise, it remains a speculation. Concerning the authors' comments: It is of course understood that the different nature of the chroma-CSF and luma-CSF applies both to HDR and SDR metrics, though the main problem I see here is in how far this defeats the idea of the paper (for SDR as well as HDR), as namely applying a known color-blind metric to color images, and in particular color defects - knowingly by a mismatch of the model and the behavior of the human visual system. Hence, in how far this discrepancy between model and HSV impacts the metric needs to be studied, probably with images that are particularly tuned. A gammut mismatch may not be good enough (this is only a "global" defect) and a lack of QP-adaption for HEVC coding may not be good enough either (this creates relatively high-frequency chroma-defects, but no defects in the frequency domain where chroma-CSF and luma-CSF differ substantially, namely at low frequencies). Hence, while the data collection and evaluation of the study are most likely correct, I am not so certain about the conclusions. They seem to me a bit premature.

Author Response

Dear reviewer,

Please find our response to your comment in the joint file

Best regards

Maxime Rousselot
